# Factors of the Readiness for Information Exchange in Pre-School Education Establishments

**Valentina Dolgova \***[ID]**, Yulia Batenova**[ID]**, Irina Emelyanova, Irina Ivanova, Lyudmila Pikuleva and Oksana Filippova**

South Ural State Humanitarian Pedagogical University, 69 Lenin prospect, 454080 Chelyabinsk, Russia
\*   Correspondence: 23a12@list.ru; Tel.: +89-193-122-570

**Abstract:** The development of an interactive environment that allows for a significant enhancement in interaction opportunities with information resources for a child is one of the primary tasks of the pre-school educational process organisation today. This study involved 50 establishments for children in Chelyabinsk ($N$ = 1280: 230 children, 350 teachers, 700 parents). We used two techniques to collect data: a questionnaire for teachers on the use of Information Exchange in the educational process, their attitude towards ICTs, and the available resources (resources included technological aspects, such as infrastructure, equipment, and software, as well as educational aspects, such as further training and the availability of a qualified methodologist); and a conversation with children on their experience in using and operating digital devices for various tasks. The questions were divided into four groups that provided us with data on the following topics: the degree of inclusion of parents in the interaction (a triad of a child, a digital device, and a parent), the awareness of the child of the capacity and resources of ICTs, motivation and interests that can be realized through digital technologies, and self-reflection as the capacity of the child to predict and analyse their activities. The Hypothesis that the level of readiness for information exchange in subjects of the educational process can be increased through further training of pre-school teachers and the establishment of a single information space in a kindergarten group has been proven right. The findings the study proved such an influence and proposed directions for further studying.

**Keywords:** pre-school children; teachers; parents; information exchange; digital devices and technologies; pre-school education establishment

---

## 1. Introduction

The world is currently undergoing remarkable changes, characterized by great dynamism and a global nature. Moreover, these changes are highly encouraged by significant shifts in the information culture, regarding the use of information and computer technologies [1–3]. In this context, cyberspace is becoming increasingly important in our lives, as it changes our human nature. This means that it induces changes in perception, consciousness, way of thinking, needs and motivations, emotional and volitional processes, activities, relationships, moral ground and values, and the vital rhythm of a human [4–6].

These changes demonstrate the necessity of theoretically and methodologically redefining the role of modern education as a social and cultural phenomenon. In the last decade, this challenge has brought the issue of a child's development in the information environment and acquisition of information technology skills to the centre of the attention of modern researchers [7–9].

The first studies on the role of computer technologies in education were published three decades ago. Today, the teaching community has to acknowledge the fact that digital usage patterns of children of all ages have changed irrevocably [10–12]. A new generation of children has appeared due to

the development of the Internet, for whom using digital technologies is natural. They utilise such technologies at home and in school, for both studying and entertainment. As their use of digital resources is growing, it will consequently have a significant effect on the rest of society. Moreover, their use of digital information sources is increasing as well. Therefore, due to a clear shift from paper-based storage to digital, children need to be able to apply modern technologies more competently and efficiently and be provided with a safe environment for virtual learning and interaction [13–15].

For example, the credibility of instructions and experts has been reduced due to their use of computer programs. Children use Google instead of reference materials and library books. They are used to searching for information independently, and as a result, their confidence in an expert or a teacher decreases. This problem raises many challenges, but the key one concerns maintaining the significance of the interaction between a child and a teacher and utilising digital technologies as a means of effective cooperation.

Pre-school children become users of the Internet at an early age. The Internet, which provides people with unlimited and free access to information, is growing constantly [16–18]. Thus, the issue of assisting children in using this information space becomes an acute and important one. However, it should be taken into consideration that although the Internet has changed the vital space of adults and is now providing them with various opportunities to enrich their lives, it has a completely different impact on children. Today, children are born into an information-intensive environment, supported by numerous digital devices. Pre-school children of the digital age live and socialise in it naturally [19–21]. As a result, they are better prepared to exist in this information-intensive environment than adults around them. This so-called digital generation can benefit from interacting with the new information environment greatly, but there are some dangers to children imposed by it [22,23]. It is well known that on 1 September 2012, a new federal act entered into force in Russia regarding the protection of children from information harmful to their health and development. This comprehensive legislation established measures to protect children against harmful information. Developing and improving information literacy and culture in children and all subjects of the educational process can be another way to avoid risks [24].

The issue of overcoming the negative psychological consequences of informatization in pre-school education, caused by computer games and animated cartoons in various forms and contents, needs to be addressed and solved scientifically. It is vital to be able to understand what happens to a child in an information environment and what changes in their personality. It is clear that teachers, psychologists, and parents can contribute to the development of information literacy in pre-school children. Today, the development of readiness for information exchange in subjects of the educational process, in accordance with the characteristics of pre-school education, is one of the primary psychological and pedagogical tasks.

As seen above, there are a few problematic areas that need further consideration:

- although the significant impact of information and communication technologies on the personality development of pre-school children is objectively recognised, such an impact is given a mixed assessment;
- considering that computer games have a strong influence on the development of pre-school children, their content is not always suitable for children of various age groups and is not always supported by either theoretical or experimental studies;
- there is a mismatch between the necessity for pre-school children to join the information environment and insufficient competence of adults with regard to assisting them in using this information environment.

The need to study information literacy comes as a consequence of the rapidly growing use of information and computer technologies (ICTs) in modern society, as it becomes crucial for people to master these skills in order to be an effective member of the information age. Information literacy is the ability to use digital technologies for research, creation, and communication for personal or

business purposes [25,26]. The notion of information literacy also covers a strong motivation to use ICTs, possession of theoretical knowledge of ICTs usage, demonstration of respective emotional and volitional qualities, and the realisation of a set of skills and competencies in the activity [27,28].

Information literacy is considered an entity, which is a complex system of connections and balances that can contribute to the development of effective conditions for information exchange [29]. Teachers face the challenge of searching for such ways of engaging children in information exchange that would ensure their safe self-realisation. However, an education establishment should not be the only responsible body in this field. The use of digital devices at home has a direct effect on the level of information literacy, and neglecting this fact can hinder the personality development of a child and their interaction with other people greatly.

By information exchange between subjects of educational process we understand exchange of information via digital means. Readiness for information exchange within the context of a pre-school establishment refers to an integrated set of psychological and operational characteristics of subjects of the educational process that is required for the successful exchange of educational and motivational information via digital technologies.

There are many computer-based, technical, and digital tools aimed at developing different mental functions in children, including visual and auditory perception, concentration, memory, and verbal reasoning. They can be effectively introduced in the education of pre-school children [30–32]. The development of an interactive environment that allows for the significant enhancement of interaction opportunities with information resources for a child is one of the primary tasks of the pre-school educational process organisation today.

All of the above determined the purpose of the study, which was to identify factors that develop the readiness for information exchange in subjects of the educational process in pre-school education establishments.

## 2. Research Methods

In this study, we compiled the results of questionnaires completed by 350 teachers from more than 50 establishments for children. This data was complemented by the results of questionnaires completed by 700 parents. We also had conversations with 230 children, aged between 5 and 8. The study was conducted in Chelyabinsk between October 2017 and April 2018.

We used two techniques to collect data: a questionnaire for teachers on the use of ICTs in the educational process in pre-school education establishments, their attitude towards ICTs, and the available resources (in this case, resources included technological aspects, such as infrastructure, equipment and software, as well as educational, such as further training and the availability of a qualified methodologist); and a conversation with children on their experience in using and operating digital devices for various tasks.

The conversation consisted of 12 questions that covered many aspects of the study. The questions were divided into four groups that provided us with data on the following topics: the degree of inclusion of parents in the interaction (a triad of a child, a digital device, and a parent), the awareness of a child of the capacity and resources of ICTs, motivation and interests that can be realised through digital technologies, and self-reflection as the capacity of a child to predict and analyse their activities. The latter served as an additional criterion.

The questionnaires were processed with the use of methods of non-numerical statistics, in particular, through the method of encoding each answer, entering the data into Excel tables and calculating the totals of the conducted quantitative analysis of the findings. Errors and missing answers were not taken into consideration, as this stage of data processing will be included in further research.

The characteristics of the children and the nature of interviewing served as the basis for sampling and were the following:

The children, aged between 5 and 8, came from average-income, complete and incomplete families.

The interviewing was performed in the form of a face-to-face conversation in the kindergarten. Answers given by the children were audio recorded and later transferred into special log sheets.

## 3. Findings

The questionnaires and conversations conducted in pre-school establishments in Chelyabinsk showed the following results. Teachers agreed upon the necessity to introduce modern information technologies into the educational process and utilise ICTs in working with parents almost unanimously (95% of them).

To interact with parents, teachers used conventional and easily accessible messengers, such as Viber and Whats App, as well as social networking sites and e-mails.

Microsoft Office and its standard components, such as Word, Excel, and Power Point, as well as various search engines, were named as the most frequently used software programs by teachers. Modern technical equipment utilised in the educational process included multimedia devices, interactive whiteboards, laptops, and printers. A small number of teachers had successfully mastered information resources such as blogging and maintaining a personal website [33,34]. In the use of information technologies, there was a division on the basis of age. While all age groups of teachers (from 23 to 50) used digital devices as a means of teaching aids, only young teachers, aged under 35, utilised communication means such as social networking sites, blogs, and personal websites (Figure 1).

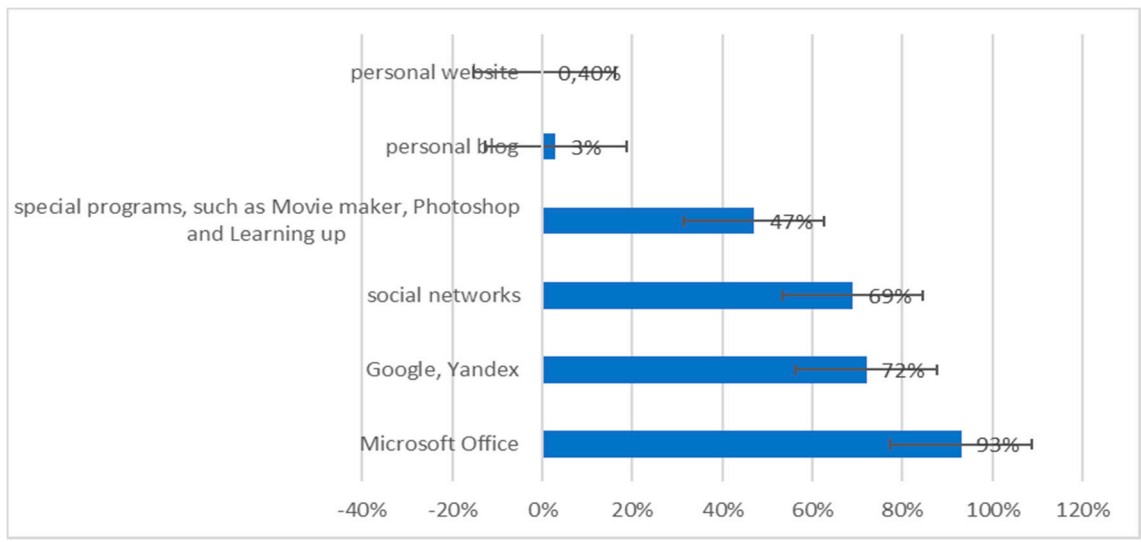

**Figure 1.** The use of information and computer technologies (ICTs) by young teachers aged under 35.

Figure 1 demonstrates the percentage of teachers who used Microsoft Office (93%) and Google and Yandex search engines (72%). A total of 3% of teachers had a personal blog, 47% used special programs, such as Movie maker, Photoshop, and Learning up, 69% utilised social networks, while only 0.4% possessed skills in creating and maintaining a personal website—however, this percentage is too low and insignificant.

When answering the question, "would you like to master new interesting programs and digital devices in order to increase the quality of your work?" all teachers responded in the affirmative. This shows their high motivation and readiness to master modern information technologies.

Next, we analysed the conversations with children on the inclusion of their parents in the process of their mastering ICTs and on their experience in using and operating digital devices for various tasks.

The first part of the conversation was regarding the engagement of children's parents in their ICTs usage. This information was collected through the following questions: "do your parents know what you do on the Internet", "do your parents help you to use a digital device", and "would you want your parents to join you and spend time with you on the Internet". Figure 2 shows the findings of this part.

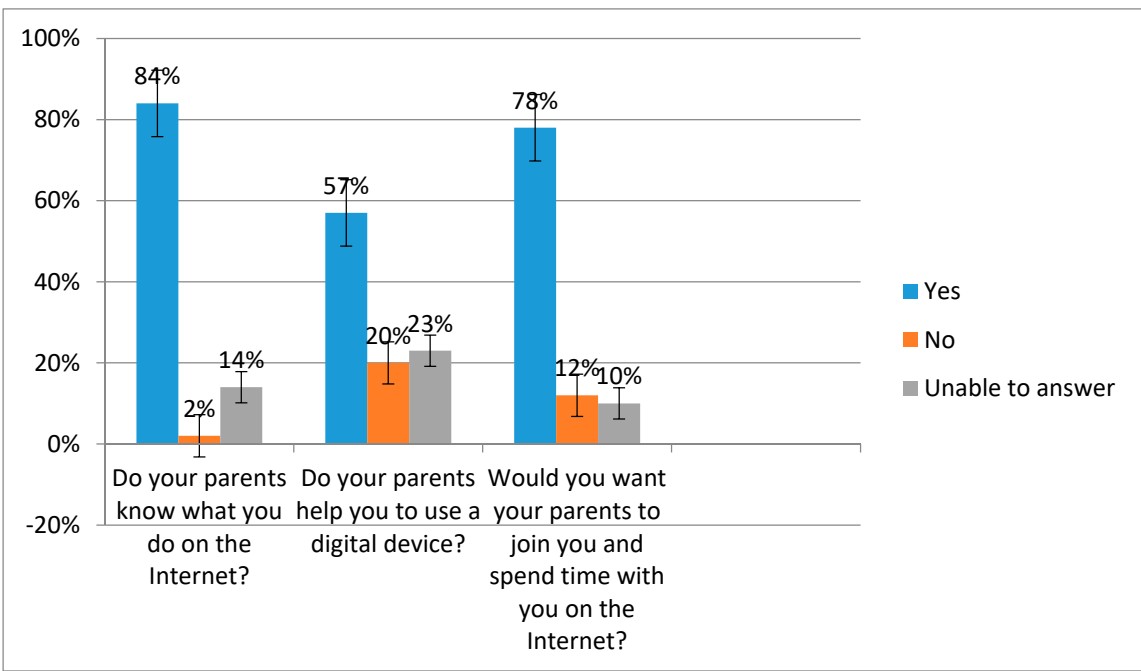

**Figure 2.** Inclusion of children's parents in the process of the children mastering ICTs.

As seen in Figure 2, 84% of children answered that their parents were aware of what they did on the Internet and what they did with a device without Internet access, 14% were unsure of the degree of inclusion of their parents, while 2% responded that their parents were totally unaware. When answering the question on their parents' assistance in their using ICTs, 57% of older pre-school children said that their parents helped them in mastering new devices and programs, 23% commented that they received little assistance from their parents, while 20% replied that their parents showed them no assistance and did not cooperate at all.

When comparing the results of this part of the parents' questionnaire and the conversations with children, it is apparent that those parents who controlled the process of digital device usage and assisted their children in using such devices fell into the category of parents who helped their children master ICTs, as noted by the children. This category included young parents as well as older ones.

The second part of the conversations with the children was regarding the awareness of the children of the capacity and resources of digital devices (Figure 3). Answers here were more diverse. The questions in this part allowed us to determine not only the degree of their mastering a technical device, but also the level of awareness of different programs and network services, or Internet-based digital technologies. When asked what devices they used, 100% of children answered with TV, 91% replied with a mobile phone, however, not all of these children owned a personal phone, which means that they actively used their parents' smartphones. This part revealed that while some parents controlled the use of the Internet by their children, others allowed them to use it on their own. The findings also showed that 34% of children owned a personal computer or a laptop, 42% had a family computer or a laptop, 37% had a tablet computer, and 27% had a video game console. As can be seen, the various devices received an almost equal distribution. However, the prevalence of TV among pre-school children is clear.

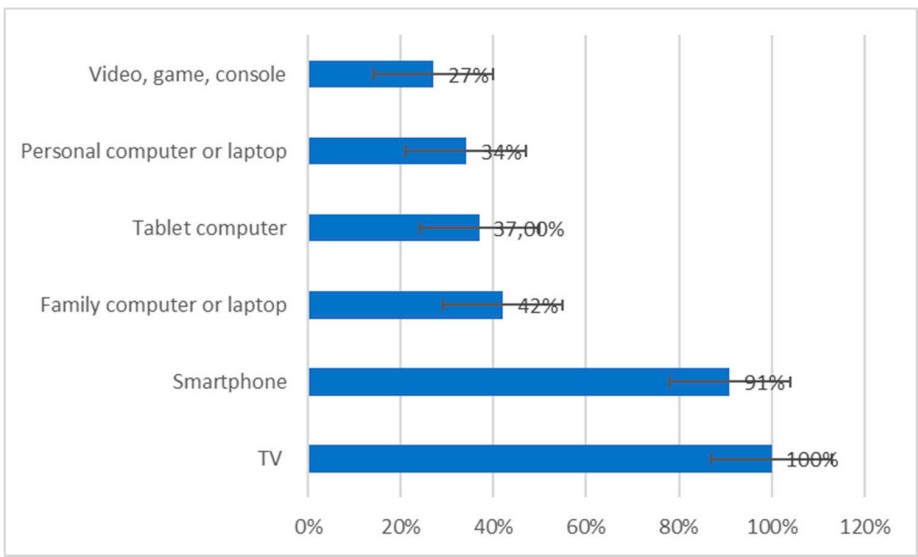

**Figure 3.** Devices used by children.

When answering the question "who taught you to use digital devices and the Internet", 33% of children said they were taught by their parents, 34% learnt how to operate devices on their own, and 18% were taught by their older siblings. Children who learnt to operate devices on their own demonstrated a higher awareness in other aspects related to the use of digital devices and the Internet. In particular, their answers to the question "what do you do on the Internet", covered a greater range of activities. A total of 100% said they played on-line and off-line games, 91% searched for interesting content (mostly pictures, videos, and music), and 64% had a personal page on a social networking site that they used for communication and uploading of personal information.

Figure 4 demonstrates the following findings: although 32% of children searched for interesting content on the Internet, no one used knowledge sites or web resources for children. These children used the Internet in order to download photos, videos, or audio content, or create their own and upload it. The video-sharing website Youtube was named the most commonly used source of information among children. A total of 23% of children remarked that they downloaded everything that was available for downloading, even if they parents did not allow it, 35% of children used devices to play games, and 18% to communicate with other gamers in on-line games or on-line communities. 38% of children said they had a personal page on a social networking site. The most common networks were Vkontakte and Moi Mir.

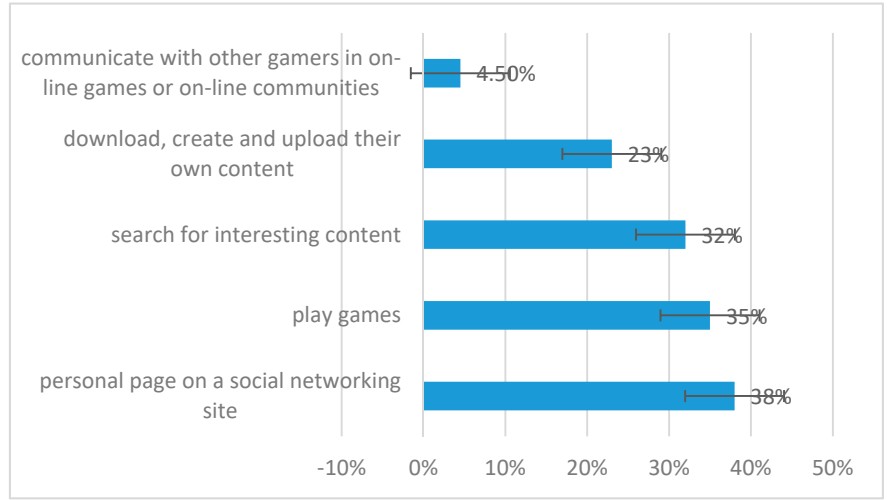

**Figure 4.** Web activities common among children.

The question "What can you do with the help of modern digital technologies?" revealed the level of information literacy and digital skills and competencies in pre-school children. Figure 5 shows the given answers and the range of acquired competencies.

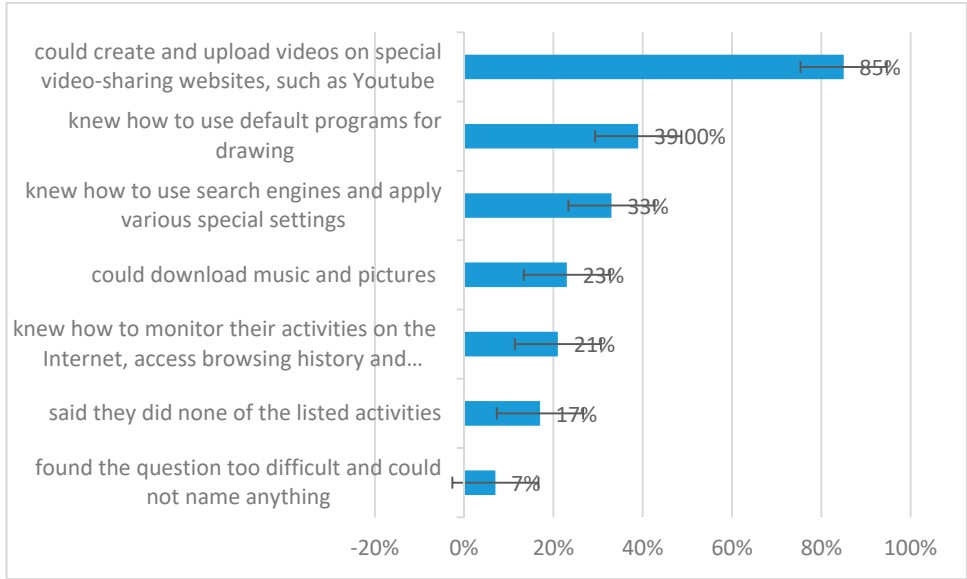

**Figure 5.** Development of skills in using digital devices in children aged between 6 and 7.

As seen in the Figure 5, 7% of children found the question too difficult and could not name anything, 17% said they did none of the listed activities, 33% knew how to use search engines and apply various special settings, 39% knew how to use default programs for drawing, 23% could download music and pictures, 8,5% could create and upload videos on special video-sharing websites, such as Youtube, 21% knew how to monitor their activities on the Internet, access browsing history and delete it, customise the home screen and organise applications.

As can be seen, one third of older pre-school children possessed basic skills in operating digital devices and were able to easily recognize the interfaces of different programs and use default programs installed in the operating system.

The third part of the conversation regarded motivation and interests. Some of the answers were partly analysed in the previous part, devoted to awareness. As seen, most of the interests of children were connected with the reception, exchange, creation, and uploading of information, as well as video games. A total of 83% of children remarked that they used ICTs for gaming. Types of video games such as shooters and racing games were the most preferred ones, as they were named by 42% of children. A total of 19.5% of the children were engaged in playing games only together with their parents—they used educational applications for children. A total of 23% of the children played on-line games for children and adults. Although 17% of children said they did not play video and computer games at all, they still had an access to smartphones and mobile gaming applications.

The additional part of the conversation was regarding the level of self-reflection in the children, in particular their evaluation of the benefits received from the use of information and communication technologies or the harm caused by it. A large percentage of the children (44%) ignored the question "what benefits can you take from the Internet". A relatively small percentage of children (ranging between 6% and 14%) answered that it was possible to use the Internet for gaming, listening to favourite music, watching cartoons, and on-line shopping.

Next, children were asked to complete the sentence, "if I had no computer and no Internet in my life, I would". However, some children did not fully understand the task, as 15% replied they would simply buy a computer or use a phone. In this case, they showed the intention to replace a missing device. On the other hand, 56% of children said they would spend more time outside, play with

toys, watch TV (it needs to be noted that watching TV takes up a great portion of children's time), draw, ride a bicycle, attend classes of different types, or build LEGO construction toys. In this case, they showed the intention to replace an activity. Only 8.5% remarked that nothing would change. Some of the answers were deeply pessimistic, as 2.5% said they would die or would not be able to live any more, and 6% answered they would be very bored or would cry.

Considering a generally positive approach and higher motivation towards mastering information and communication technologies [35,36] among teachers, the necessity of further training in the field needs to be acknowledged. Unfortunately, as of today, we do not possess any credible data on the level of information preparedness in teachers. Therefore, we are not able to propose any improved requirements for the digital skills development in the process of information exchange. However, it is possible to develop and present some educational programs that include mastering of information-related skills by teachers and children, as well as improving the cognitive and the value-motivational spheres that serve as indicators of the level of readiness for information exchange.

There is a tendency today that demonstrates the necessity of teachers to increase their level of technological literacy due to improvements made in resources in the education system [37,38]. It means that those teachers who have mastered the use of ICTs in pre-school establishments on their own or through specialized courses will be gradually replaced by teachers who have studied information technologies in the framework of their training. However, the possession of technological literacy should not prevail over such psychological aspects of readiness as cognitive, emotional, value-motivational, and communicative qualities [39,40]. It is clear that simply placing digital devices in education establishments for children will not improve the quality of education and knowledge acquisition. The key to understanding principles of effective information exchange is the combination of technical capacities, pedagogical conditions, and psychological readiness.

A teacher must master the following aspects: (1) the primary approaches towards the use of digital devices and computer programs in the educational process; (2) the stages of developing a computer program for children; (3) ergonomic factors in designing a user system for pre-school children; (4) the user interface elements required from a computer program for children; (5) the parameters of information organisation and arrangement of visual components; (6) methodological materials for educational computer programs for children; and (7) the differences between commercial computer games and games developed by experts for pre-school children.

## 4. Conclusions

It should be noted that the issue of benefits received from digital technologies in the sphere of education remains understudied. However, the theory that the level of readiness for information exchange in subjects of the educational process can be increased through further training of pre-school teachers and establishment of a single information space in a kindergarten group has been proven right. Very few people still doubt the influence of digital technologies on our everyday life. The study findings proved such an influence and proposed directions for further studying.

**Author Contributions:** conceptualization, V.D., Y.B., I.E., I.I., L.P., O.F.; methodology, V.D., Y.B., I.E., I.I., L.P., O.F.; software, V.D., Y.B.; validation, V.D., Y.B., I.E., I.I., L.P., O.F.; formal analysis, V.D., Y.B., I.E., I.I., L.P., O.F.; investigation, V.D., Y.B., I.E., I.I., L.P., O.F.; resources, V.D., Y.B., I.E., I.I., L.P., O.F.; data curation, V.D., Y.B., I.E.; writing—original draft preparation, V.D., Y.B., I.E., I.I., L.P., O.F.; writing—review and editing, V.D. visualization, V.D., Y.B., I.E., I.I., L.P., O.F.; supervision, V.D., Y.B.; project administration, V.D., funding acquisition, V.D., Y.B., I.E., I.I., L.P., O.F.

**Funding:** This work was supported by the Russian Foundation for Basic Research Grant No. 18-013-00743 A "Establishing the Basics of Information Literacy for Preschoolers".

**Acknowledgments:** The article is written in the framework of the Scientific and Methodological Foundations of Psychology and Management Technology of Innovative Educational Processes in the Changing World scientific project of the comprehensive plan of research, project and organizational activities of the Research Centre of Russian Academy of Education in the South Ural State Humanitarian Pedagogical University for 2018–2020.

**Conflicts of Interest:** The authors declare no conflicts of interest.

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
