# Peer review of "Factors of the Readiness for Information Exchange in Pre-School Education Establishments"

_education, doi:10.3390/educsci9030166_

Round 1
Reviewer 1 Report
Overall, the study significantly contributes to the field of technology education to the kindergarten and lower primary students. Except for minor missing information in the method session and some minor spelling mistakes, this paper would be a high interest to the academia as well as to the educational practices.
In figure 1, author must clearly specify the legend for each of the bar of the graph. i.e “special” is a vague term. Figure 2’s title is unclear. Figure 3 graphic is not suitable, author could use a circle chart instead.
Author must identify the children subject in more details: education level, social-economic background in the method session. Author must elaborate on the data analysis: what tools did the use to analyze data? How missing data and error analysis were identified and the method of eliminate such data as well as bias data. Also, author must elaborate on how the interviews with children were conducted to ensure the ethical aspect if human subject.
Should address some spelling mistake (i.e. figure 5)
Line 161. The interview question is not objective, as it is “affective”
instead of neutral. This yes-no type of question is a high potential for bias
data. Hence, it should be eliminated from the paper.
Author Response
Dear Reviewer!
Good day!
I made the necessary changes.
Changed the topic to a more appropriate content of the article and removing part of the comments of the first distinguished reviewer.
Changed the composition of prickly words.
Changed all the charts. Indicated errors.
Thank you very much for working with our article.
We hope for the publication. We really hope! ))))
Best regards, Prof. Dolgova Valentina,
Doctor of Sciences (Psychology), Dean of the Faculty of Psychology, South Ural State Humanitarian Pedagogical University, Chelyabinsk, Russia.

Reviewer 2 Report
This is a work of interest in the area that is framed, treated with adequate rigor in both the methodology, and in the analysis of results, and obtains relevant conclusions that it discusses in an appropriate manner in the last section.
Author Response

(The authors gave the same response as above.)
